# *Lactobacillus plantarum* Alters Gut Microbiota and Metabolites Composition to Improve High Starch Metabolism in *Megalobrama amblycephala*

**DOI:** 10.3390/ani15040583

**Published:** 2025-02-18

**Authors:** Linjie Qian, Siyue Lu, Wenqiang Jiang, Qiaoqiao Mu, Yan Lin, Zhengyan Gu, Yeyang Wu, Xianping Ge, Linghong Miao

**Affiliations:** 1Wuxi Fisheries College, Nanjing Agricultural University, Wuxi 214081, China; qianlinjiejie@gmail.com (L.Q.); qiaoqiao72021@163.com (Q.M.); 2Key Laboratory for Genetic Breeding of Aquatic Animals and Aquaculture Biology, Freshwater Fisheries Research Center, Chinese Academy of Fishery Sciences, Wuxi 214081, China; lusiyue@ffrc.cn (S.L.); jiangwenqiang@ffrc.cn (W.J.); liny@ffrc.cn (Y.L.); guzhengyan@ffrc.cn (Z.G.); 3ANYOU Biotechnology Group Co., Ltd., Taicang 215421, China; wuyeyang87@163.com

**Keywords:** *Megalobrama amblycephala*, high-starch diet, *Lactobacillus plantarum*, metabolomics, gut microbiota

## Abstract

*Lactobacillus plantarum* contributes to the improvement in digestive function and immunity in aquatic animals. *Megalobrama amblycephala* has weak tolerance to glucose. High carbohydrate levels can lead to lipid deposition, the dysregulation of glucose metabolism, and even immune resistance. Adding *Lactobacillus plantarum* could effectively improve the gut microbiota disorder caused by high starch levels, improve glycolipid metabolism and bile acid metabolism, reduce intestinal cholesterol absorption, and improve immunity, thereby protecting the liver to a certain extent.

## 1. Introduction

Probiotics are living microorganisms that can be metabolized into a wide range of nutrients and micronutrients such as vitamins, fatty acids, and essential amino acids to support the growth and health of aquatic animals [1]. Over the past decade, the use of probiotics has emerged as the dominant method for maintaining the balance of the mammalian gut microbiota [2]. Probiotics have physiological functions such as improving intestinal health, promoting digestion and absorption, participating in immune regulation, relieving allergies, having anti-tumor effects, and lowering serum cholesterol [3,4]. Consuming a diet supplemented with yeast culture improved the growth performance, feed utilization, and hepatic glucose metabolism of *Micropterus salmoides* [5]. The gut microbiota is a highly dynamic system whose abundance and composition are noticeably affected by various exogenous factors, including diet composition [6], life stage [7], and antibiotic intake [8]. The gut microbiota produces neurotransmitters, short-chain fatty acids (SCFAs) [9], branched amino acids [6], cholecystokinin [10], and glucagon-like peptide [11]. Bile acid (BAs) metabolism in the intestine and liver is vital for glucose and lipid metabolism regulation. In nonalcoholic fatty liver disease (NAFLD), supplementing rats with 312 mg/kg/day of *Eosinophil-Lactobacillus* was found to decrease blood lipid and total BA levels while activating farnesoid X receptor (*Fxr*) and fibroblast growth factor 15 (*Tgf15*) in the liver. Additionally, it exhibited a regulatory effect on intestinal microbial diversity by promoting increased beneficial bacteria abundance while suppressing pathogenic microbial populations [12]. Similarly, in type II diabetic (T2DM) rats, the addition of *Ampelopsis grossedentata* was shown to increase the activity levels of the intestinal bile salt hydrolase-active gut microbiota and activate the expression of *Fxr* and *Tgf15* in the liver–intestine to ameliorate the disorders of glucolipid metabolism [13].

Lactic acid is the main metabolite of *Lactobacilli* present during glucose fermentation [14]. *Lactobacilli* can degrade intestinal toxin receptors, maintain normal intestinal pH, increase intestinal motility, and maintain intestinal permeability integrity [15]. Most probiotics require a sufficient number of live bacteria to be effective, and live bacteria are superior to heat-inactivated bacteria for the competition and displacement inhibition of adherent IPEC-JC cells [16]. The beneficial effects of probiotics may be facilitated by their ability to attach to the gut, promoting intestinal colonization and prolonging its persistence [17]. *Lactobacillus* utilizes intestinal epithelial surface components and metabolites to induce mucin expression, enhance intestinal epithelial tight junctions, and protect intestinal epithelial cells [18]. *Lactobacillus plantarum* (LAB) has been shown to have proteins associated with biofilm formation that were able to recognize glucose and mediate adhesion to human colonocytes [19]. LAB had a therapeutic effect on colitis in mice by modulating the intestinal microbiota and increasing SCFAs levels to attenuate the inflammatory response and protect the mucosal barrier [20]. LAB significantly regulated glycerophospholipid metabolism, fatty acid degradation, and lipid metabolism and promoted the metabolic efflux of BAs in NAFLD mice [21]. Numerous studies on aquatic animals have reported the effects of the dietary supplementation of *Lactobacillus* on improving liver lipid deposition and growth performance, including *Megalobrama amblycephala* (*M. amblycephala*) [22], *Oreochromis niloticus* [23], and *Litopenaeus vannamei* [24]. Limited research has been conducted to investigate the regulatory role of LAB in nutrient absorption and metabolism, such as the interaction between glucose metabolism and BAs metabolism through modulating gut microbiota and metabolites.

*M. amblycephala*, a freshwater farming species, is susceptible to the high level of dietary carbohydrates, 30% carbohydrate level was suitable for 15 g juvenile *M. amblycephala* [25]. High carbohydrates may lead to lipid deposition, dysregulation of glycolipid metabolism, and even immune resistance [26]. In previous studies, we found that diet supplementation of LAB improved the utilization of the high starch diet and the growth performance of *M. amblycephala* [22]. However, the mechanism of how LAB promote utilizing dietary starch through regulating gut function is still unclear. We further investigated the key gut microbiota and metabolites that regulate high starch metabolism, and demonstrated how LAB affects the regulatory targets of high starch metabolism through the intestine, which will provide an important basis for the healthy breeding of *M. amblycephala*.

## 2. Materials and Methods

### 2.1. Ethical Statement

According to the guidelines of the Animal Care Advisory Committee of the Chinese Academy of Fishery Sciences (Authorization No. 20200903001), all experiments and research activities involving animals must stringently follow the relevant protocols and standards of practice to ensure that the rights and interests of animals are fully protected.

### 2.2. Experimental Diets

A basic diet (Diet LW, including 15% wheat starch) and a high starch diet (Diet HW, including 30% wheat starch) were prepared based on the nutrition requirements of juvenile *M. amblycephala* (Table 1). The nutrient composition of feed was determined according to the national standard. Dry matter content was calculated by atmospheric pressure drying method at 105 °C (GB/T6435-2014) [27], and crude protein content was determined by Kjeldahl nitrogen determination method (GB/T6432-2018) [28]. The crude fat content was measured with Soxhlet extraction method (GB/T6433-2006) [29], and the crude ash content was measured with 550 °C (GB/T6438-1992) [30].

### 2.3. The Addition of LAB

LAB strains were isolated and provided by Jiangsu Suwei Biological Research Co., Ltd. (Wuxi, China). After dissolution in 0.9% sterilized saline, LAB solution was evenly sprayed into HW feed and then air-dried (Diet HP, including LAB in HW feed). The air-dried feed were kept at −20 °C for storage. The HP feed were prepared every week, and the concentration of LAB was tested in each batch. The concentration of LAB in the HP feed was kept at 10^6^ cfu/g as measured by plate count method.

### 2.4. Fish and Feeding Management

*M. amblycephala* were obtained from the national stock farm (Wuhan, China). The juveniles were allowed to acclimate to the farming environment for one week and temporarily fed with commercial feed (Tongwei Co., Ltd., Wuxi, China). The feeding trial was conducted in Nanquan farm of Freshwater Fisheries Research Center, Chinese Academy of Fishery Sciences (120.29 E, 31.43 N). A total of 180 healthy and uniform fish (initial average weight 13.5 ± 0.5 g) were randomly distributed to 9 outdoor floating cages (1 m × 1 m × 1 m) at the density of 20 fish per cage. Either one experimental diet was appointed to 3 floating cages randomly. The fish was satiated fed 3 times a day (8:00, 12:00, and 17:00) for 8 weeks. The daily intake of feed was 5% of their body weight. During the 8-week feeding experiment, the water temperature was kept between 28 and 31 °C, dissolved oxygen ≥ 7 mg/L, ammonia nitrogen ≤ 0.1 mg/L, pH 7.3–7.8. The natural light cycle (12 L:12 D) was adopted.

### 2.5. Sample Collection

After the 8-week feeding experiment, fish were fasted for 24 h before sampling. Six fish were randomly selected from each floating cage and immediately anesthetized with 100 mg/L MS-222. The liver and intestine were quickly collected. Part of the liver and intestine were fixed with 2.5% glutaraldehyde for transmission electron microscope analysis. The remaining part were flash-freezed in liquid nitrogen, then transferred and stored at −80 °C for subsequent gene expression determination. The intestinal contents from the hindgut were collected for metabolites analysis, and then the intestinal mucosa were collected for gut microbiota analysis.

### 2.6. Gut Microbiota Analysis

Microbial DNA was extracted from hindgut samples using the E.Z.N.A.^®^ DNA Kit (Omega Bio-tek, Norcross, GA, USA). The V4-V5 region of the bacteria 16S ribosomal RNA gene was amplified by PCR (95 °C for 2 min, followed by 25 cycles at 95 °C for 30 s, 55 °C for 30 s, and 72 °C for 30 s and a final extension at 72 °C for 5min) using primers 515F (5′-barcode-GTGCCAGCMGCCGCGG)-3′ and 907R (5′-CCGTCAATTCMTTTRAGTTT-3′). Finally, the amplified PCR products were sent to Illumina Miseq platform (Shanghai BIOZERON Co., Ltd., Shanghai, China) for high-throughput sequencing [31].

### 2.7. Metabolites Analysis

An amount of 100 mg of metabolite sample ground in liquid nitrogen was prepared and 500 μL of 80% methanol in water was added. The sample was vortexed and put on ice for 5 min and centrifuged at 15,000× *g* for 20 min at 4 °C. A certain amount of supernatant was collected and diluted with mass spectrometry-grade water until the methanol content was 53%, and then centrifuged at 15,000× *g* for 20 min at 4 °C. In addition, the supernatant was collected, and the samples were analyzed using LC-MS [32].

Chromatographic separation was performed on an ACQUITY UPLC HSS T3 column (100 mm × 2.1 mm, 1.7 μm, Waters, Milford, MA, USA) at a column temperature of 40 °C and a flow rate of 0.35 mL/min, where the A mobile phase consisted of water and 0.1% formic acid and the B mobile phase was acetonitrile [33]. The metabolites were eluted using the following gradient: 0–1.0 min, 5% B; 1.0–9.0 min, 5–100% B; 9.0–12.0 min, 1000% B; and 12.0–15.0 min 5% B. The volume of each sample was 5 μL.

### 2.8. Real-Time PCR Analysis on Gene mRNA Expressions

Total RNA was extracted from the liver and intestine by using RNAisoplus kit (Takara, Dalian, China). The quality of extracted total RNA was detected by Nano Drop2000 (Thermo Fisher Scientific Inc., Waltham, MA, USA). Primer sequences of *fxr*, low-density lipoprotein (*ldlr*), liver X receptor (*lxr*), cholesterol 7α-hydroxylase (*cyp7a1*), vitamin α X receptor (*rxr*), 3-hydroxyl-3-meglutaryl-CoA reductase (*hmgcr*), liver nuclear factor 4α (*hnf4α*), ATP binding box transporter A1 (*abca1*), nuclear receptor subfamily 0 group B member 2 (*shp*), sterol regulatory element-binding transcription factor1c (*srebp1c*), pyruvate kinase (*pk*), glucose-6-phosphatase (*g6pase*), acetyl-CoA carboxylase (*acc*), adipose synthase (*fas*), lipoprotein lipase (*lpl*), peroxisome proliferator-activated receptor delta (*pparβ*), and *β-actin* are shown in Table 2. *β-actin* was used as the reference gene. All primers were synthesized by Shanghai Shenggong Bioengineering Co., Ltd. (Shanghai, China). The relative expressions of the target genes were calculated using the 2^−ΔΔCt^ method.

### 2.9. Transmission Electron Microscope Analysis

Firstly, the pre-fixed intestinal samples were rinsed with phosphate-buffer solution (PBS) four times repeatedly to remove the excess fixative. Then, the samples were fixed in 1% osmium tetroxide solution for 2 h to effectively preserve the ultrastructural morphology of the cells. After fixation, the samples were dehydrated by gradient acetone dehydration. The specific steps were as follows: soak the samples once in 30%, 50%, 70%, and 90% acetone solution, and finally soak the samples three times in 100% acetone solution to maximally protect the three-dimensional structure of the samples. Subsequently, the dehydrated samples were immersed in a mixture of acetone: resin = 1:1 for 24 h to allow the resin to completely penetrate into the samples. Finally, the samples were transferred to the pure resin solution and cured at 70 °C for 24 h. After curing, the embedded samples were sliced at a thickness of 60 nm by using ultramicrotome (UC 7, Leica, Weztlar, Germany). After sectioning, the samples were double-stained with uranyl acetate-lead citrate. Finally, the stained sections were placed under transmission electron microscope (H-7500, Hitachi, Tokyo, Japan) for observation and photographs.

### 2.10. Statistical Analysis of Data

The data were analyzed using SPSS 25.0 software (SPSS Inc., Chicago, IL, USA) after confirming the assumptions of normality and homogeneity of variance. An independent *t* test was used to compare the differences in gut microbiota and metabolites between every two groups. Duncan’s test was employed to analyze the relative expression of genes among different treatments. Data were presented as the mean ± SEM with the differences were considered statistically significant at *p* < 0.05. Advanced Cor link was performed using the OmicStudio tools at https://www.omicstudio.cn/tool accessed on 14 May 2024.

## 3. Results

### 3.1. Effect of Lactobacillus plantarum on Gut Microbiota of Juvenile M. amblycephala

The 16S rRNA amplification and sequencing were conducted to determine the contributions of individual groups of gut microbiota (Figure 1). A total of 5967 OTUs were detected in all three groups (Figure 1A). PCA results showed a significant separation between LW and HP groups, with 23.62% in PCA2 and 8.39% in PCA5 (Figure 1B). Chao1 (*p* = 0.003) (Figure 1C) and Shannon (*p* = 0.009) (Figure 1D) indexes in the HP group were remarkably decreased than in the LW group. Nevertheless, Simpson (*p* = 0.039) (Figure 1E) showed the opposite. The top three dominant gut microbiota at the phylum level were Proteobacteria (31.04%), Desulfobacterota (12.41%), and Actinobacteriota (11.31%) in the LW group, while Proteobacteria (38.03%), Actinobacteriota (8.54%), and Firmicutes (7.34%) in the HW group, and Proteobacteria (47.64%), Firmicutes (11.89%), and Cyanobacteria (7.94%) in the HP group (Figure 1F). At the genus level, the top three dominant gut microbiota were *Chloroplast* (4.60%), *Desulfocapsaceae* (3.23%), and *Woeseia* (3.20%) in the LW group, *Ralstonia* (9.14%), *Xanthobacteraceae* (1.76%), and *Chloroplast* (1.42%) in the HW group, and *Ralstonia* (27.99%), *Cetobacterium* (3.27%), and *ZOR0006* (3.25%) in the HP group (Figure 1G).

The relative abundance of Acidobacteriota (*p* < 0.001), Actinobacteriota (*p* = 0.001) and Desulfobacterota (*Desulfocapsaceae*) (*p* = 0.008, *p* = 0.006) (Figure 2C,I *Desulfocapsaceae*) were obviously reduced in the HP group compared with the LW group. The relative abundance of Fusobacteriota (*Cetobacterium*) (*p* = 0.040, *p* = 0.035) (Figure 2D,H *Cetobacterium*) was decreased in the LW group compared to the other two groups. The relative abundance of Firmicutes (*ZOR0006*) (*p* = 0.006, *p* = 0.039) (Figure 2E,G *ZOR0006*) in the HP group was remarkably higher than in the LW group. The relative abundance ratios of Firmicutes and Bacteroidota (Figure 2F) were significantly higher in HW (*p* = 0.043) and HP (*p* < 0.001) groups than in the LW group (Figure 2F). As for Proteobacteria, the relative abundance of *Ralstonia* (*p* = 0.035) (Figure 2J) and *Caulobacteraceae* (*p* = 0.029) (Figure 2L) was remarkably higher in the HP group than in the LW group. The relative abundance of *Woeseia* (Figure 2K) in the LW group was extremely higher than HW (*p* = 0.007) and HP (*p* = 0.020) groups. Compared with the HW group, the relative abundance of *Xanthobacteraceae* (*p* = 0.041) (Figure 2M) was obviously lower in the HP group.

### 3.2. Effect of Lactobacillus plantarum on Metabolites of Juvenile M. amblycephala

Under the positive ion mode, the count of metabolites illustrated in the KEGG database was 365 (Figure 3A), and under the negative ion mode, the count of metabolites illustrated in the KEGG database was 636 (Figure 3B). There were 39 differential metabolites in the LW and HW groups (Appendix A), 73 differential metabolites in the LW and HP groups (Appendix A), and 36 differential metabolites in the HW and HP groups (Appendix A). Metabolite Venn showed that LW group vs HW group had 13 co−differential metabolites with LW group vs HP group and 2 co-differential metabolites with HW group vs HP group. 11 co-differential metabolites were shared between LW group vs HP group and HW group vs HP group (Figure 3C). PCA showed that the HP group was separated from LW and HW groups, with 78.13% in PC1, 2.30% in PC2 (Figure 3D).

The 26 differential metabolites were divided into 8 categories, of which 11 metabolites were classified as Glycerophospholipids, 3 as Fatty acyls, and 2 as Pteridines and derivatives (Figure 4A, Appendix A). The HW group had obviously lower levels of fatty acid amide hydrolase inhibitor (JNJ–1661010) (*p* < 0.001), 5–Methyltetrahydrofolic acid (*p* = 0.019) and 2′–O–Methyladenosine (*p* = 0.028) (Figure 4C,D,F) than the LW group. Whereas phosphatidylcholine (PC) (16:1/20:5) (*p* = 0.022), Lysophosphatidylcholine (LPC) 22:6 (*p* = 0.026), Lysophosphatidylserine (LPS) 20:1 (*p* = 0.023), and Taurochenodeoxycholic acid (TCDCA) (*p* = 0.030) (Figure 4H–J,L) in the HP group were obviously lower than those in the HW group.

### 3.3. Effect of Lactobacillus plantarum on Liver and Intestinal Bile Acids and Glycolipid Metabolism of Juvenile M. amblycephala

In the intestine, the relative expressions of *fxr* (*p* = 0.021), *hmgcr* (*p* < 0.001), *hnf4α* (*p* = 0.007), *rxr* (*p* = 0.018), and *shp* (*p* = 0.012) (Figure 5A,D,E,G,H) were significantly higher in the HP group than those in the LW and HW groups. The relative expressions of *acc* (*p* = 0.004), *fas* (*p* = 0.014), *lpl* (*p* < 0.001), *ldlr* (*p* = 0.019), and *pk* (*p* = 0.042) (Figure 6A–C,E,F) in the HP group were remarkably higher than those in the other two groups. However, the relative expressions of *g6pase* (*p* < 0.001) and *pparβ* (*p* < 0.001) (Figure 6G,H) in the LW group were obviously higher than those in the other two groups.

In the liver, the relative expressions of *cyp7a1* (*p* = 0.005) (Figure 5J) in HP group were remarkably lower than those in the LW group. The relative expression of *lxr* (*p* = 0.018) (Figure 5N) in the HP group was significantly lower than in the HW group. However, compared with the LW group, *abca1* (*p* = 0.010) (Figure 5K) had a significant rise in the HP group. The relative expressions of *lpl* (*p* = 0.045) and *ldlr* (*p* = 0.005) (Figure 6K,M) in the HP group were obviously lower than those in the HW group (*p* < 0.05), while the relative expressions of *srebp1c* (*p* = 0.047) and *pparβ* (*p* = 0.005) (Figure 6L,O) were significantly lower than those in the LW group.

### 3.4. Correlation Analyses Combined with Gut Microbiota, Metabolites, and Gene Expressiones

Firmicutes was positively associated with *Cetobacterium*. *Cetobacterium* was positively associated with *fxr I* and *hnf4α I*, Neopterin was positively correlated with *rxr I*, and S–Lactoylglutathione was positively correlated with *g6pase I* (*p* < 0.05). PC (4:0/18:5), LPS20:1, and LPC22:6 were negatively correlated with *rxr I* (*p* < 0.05). JNJ–1661010 was positively correlated with *pparβ I* and *g6pase I*, while ACar20:1 was the opposite (*p* < 0.05). In the liver, PC (16:1/20:5) was significantly correlated with *ldlr L* (*p* < 0.05) (Figure 7).

### 3.5. Effect of Lactobacillus plantarum on the Histopathology of the Intestine and Liver

The nucleus of the cells in the HW group was significantly deformed in the liver, which had more lipid droplets (Figure 8A). Compared with the HW group, there was a normal morphology of the nuclei (Figure 8B) and fewer lipid droplets in the HP group. The intestine of the HW group appeared partially fused with mucus and slightly swollen at the tip (Figure 8C). On the contrary, autophagy was observed in the HP group (Figure 8D).

## 4. Discussion

The gut microbiota was primarily composed of symbionts, collectively known as the gut microbiota. Due to differences in physiology and nutritional intake, each host had its own specific core of gut microbes. Feed composition was one of the crucial factors that affect the gut microbes of animals. Reduced gut microbial diversity in mice fed a high-fat diet was effectively alleviated by the addition of tea polyphenols [34]. In this experiment, the gut microbiota diversity was significantly reduced while LAB was supplemented in HW diet. Similarly, the addition of LAB reduced the diversity and richness of the gut microbes in turbot [35]. The addition of *Lactobacillus johnsonii* and blueberry extracts to high fat diets significantly decreased “vitamin alpha diversity” [36]. In dairy cows, the *Lactobacillus casei* Zhang and *Lactobacillus plantarum* P-8 treatment groups were found to significantly increase the abundance of beneficial bacteria but suppress some conditionally pathogenic bacteria [37]. The use of *Bifidobacterium bifidum* to treat *Helicobacter pylori* -positive patients showed a dramatic decrease in alpha diversity and a reduction in *Helicobacter pylori* [38]. Therefore, we speculated that consuming adequate amounts of probiotics may lead to competitive elimination of pathogenic bacterial strains.

In mammals, energy deposition was related to the distribution of intestinal bacteria [39]. More efficient absorption and utilization of food calories would be facilitated when there were more Firmicutes than Bacteroidetes in the intestine [40]. The majority of research suggested that higher ratios of them may lead to obesity [41,42]. In this experiment, the ratio of Firmicutes and Bacteroidetes significantly raised in the HP group, indicating that LAB enabled better digestion of starch in feed and promoted the growth of *M. amblycephala* which was consistent with our growth results (Appendix A). Similarly, the increase in *Cetobacterium* also confirmed this conjecture. Researches reported that *Cetobacterium* was a component of the freshwater fish microbiota. It was also able to produce SCFAs, enhanced insulin secretion and raised the body’s glucose utilization capacity [43]. Litchi chinensis regulated insulin resistance and glucose metabolism and improve zebrafish obesity by modulating *Cetobacterium* [44]. Firmicutes was shown to produce secondary BAs [45], which was found to have the physiological function of promoting lipid digestion and absorption. The addition of *Pleurotus eryngii* polysaccharide to mice on high lipid diet could increase the excretion of BAs and lipids as well as the relative abundance of Firmicutes, thus preventing obesity in obese mice [46]. BAs were synthesized from cholesterol in the liver and enter the intestinal tract through the bile duct, reabsorbed at the distal end of the intestinal tract. The reabsorbed BAs were circulated back to the liver through peripheral circulation and re-metabolized and secreted to establish their enterohepatic circulation [47]. Transmission electron microscopy results showed that the addition of LAB to high starch diets remarkably reduced the area of lipid droplets in the liver and reduced hepatic lipid deposition. So, we speculated that LAB could activate cholesterol metabolism to produce bile acids by increasing the ratio of Firmicutes/Bacteroidetes and *Cetobacterium*, thereby stimulating the utilization of starch. Similarly, gene results confirmed this speculation. BAs in the intestine activated *FXR*/*RXRα* and thus regulated bile acid synthesis [48]. High glucose induced oxidative stress in cardiomyocytes negatively regulated *RXR* expression [49]. *Fxr* regulated multiple metabolisms, especially playing vital roles on bile acid metabolism [50]. Activation of intestinal *fxr* can restore hepatic homeostasis [51]. *FXR* could indirectly sense glucose concentration and synergistically translate bile acid and glucose-feeding signals into enhanced transcriptional output [52]. LAB H-87 was found to activate the FXR signaling pathway to inhibit hepatic fat deposition in mice [53]. In this experiment, the addition of LAB to a high starch diet activated the expression of *fxr* and *rxr* in the intestine and initiated endogenous cholesterol synthesis.

Armeni et al. [54] had demonstrated that S-Lactoylglutathione could be used as an alternative source of mitochondrial glutathione. In T2DM mice, the increase in S-Lactoylglutathione was found to increase glycolysis and convert most of the glucose to pyruvate-lactate [55]. Our research found that compared to the HW group, the S-Lactoylglutathione content was increased in the HP group and positively correlated with the expression of intestinal *g6pase*. The expression of *G6pase*, which were key enzymes in the gluconeogenesis pathway, were significantly reduced in mice that fed probiotic-fermented milk [56]. We also found that LAB in the gut significantly downregulated the expression of *pparβ* and *g6pase* and activated glycolysis. Similarly, *Lactobacillus casei* could dramatically downregulate the relative expression of *G6pase* in the liver of T2DM mice [57]. *HNF4α* regulated glycolytic enzymes and glucose transporters as well as lipid homeostasis [58]. *SHP* was part of a negative feedback loop in the transcriptional regulation of genes involved in multiple metabolic pathways, including bile acid and glucose homeostasis, which is negatively regulated by *PI3K* [59,60]. The expansion in the expression of the cholesterol absorption gene *shp* could promote the absorption of cholesterol in the intestine. Interestingly, this result was consistent with the conjecture above, which verified the hypothesis that LAB can promote intestinal cholesterol absorption. Mitochondrial function has been closely associated with oxidative stress, and the damage to mitochondria exacerbates oxidative stress and inflammation [61]. Intestinal transmission electron microscope showed that high starch caused mitochondrial swelling, whereas the addition of LAB alleviated this symptom and the occurrence of autophagy. In ulcerative colitis, cells cleared damaged mitochondria and overloaded reactive oxygen species by mitochondrial autophagy, which protected the intestinal trac [62]. Li et al. found that *Lactobacillus acidophilus* could promote autophagy and inhibit inflammation in mice [63]. Neopterin activated human helper T cell type 1 immunity [64]. A previous study indicated that LAB was able to improve the antioxidant capacity of the liver of *M. amblycephala.* (Appendix A). Consequently, we hypothesized that LAB could effectively ameliorate high starch-induced mitochondrial damage and improve the antioxidant capacity of *M. amblycephala*.

The most abundant phospholipid component of lipid droplets was PCs [65]. In the mouse model of insulin resistance, the regulation of bile acid metabolism and glucose homeostasis were found to be dependent on the PC signaling pathway [66]. With the addition of LAB to high starch diet, the amount of PC (16:1/20:5) in *M. amblycephala* were reduced. Likewise, transmission electron microscope and oil red O staining (Appendix A) showed that the area of lipid droplets was significantly reduced in the HP group. The increase in TCDCA was associated with high insulin resistance [67]. High levels of TCDCA could cause cholestasis, and probiotic supplementation was found to alleviate PFBS-induced metabolic disorders in zebrafish [68]. This resembled the results of our research. The SCFAs generated by glycolysis activate lipolysis and reduce lipid deposition in liver. *LXR* mainly functions as a reverse cholesterol transporter in lipid metabolism. The activation of *LXR* promotes the reverse transport of cholesterol to the liver and accelerates the conversion of cholesterol to BAs in the live [69]. LAB KLDS 1.0344 was found to reduce cholesterol by upregulating intestinal *LXR* expression [70]. In this study, a high starch diet activated the expression of *ldlr*, *hnf4α*, and *lxr* in the liver. It was speculated that a high starch diet promotes the reverse transport of cholesterol, which may lead to the burden of cholesterol in the liver. However, the addition of LAB in high starch diet significantly activated the expression of *abca1* in the liver. In mice, *Abca1* deficiency was found to accumulate cholesteryl esters in tissues, increasing the risk of cardiovascular disease [71]. This finding further suggested that the intake of LAB promotes the bile acid cycle in the liver, absorption, and decomposition due to high starch intake in the diet.

## 5. Conclusions

In conclusion, supplementation with *Lactobacillus plantarum* could effectively improve the gut microbiota disorders caused by high starch, activated intestinal glycolysis, increased intestinal cholesterol absorption, stimulated the bile acid cycle, and generated mitochondrial autophagy, thereby protecting the liver to a certain extent. These findings suggested that the modulation of gut bacterial microbiota with *Lactobacillus plantarum* can bring benefits to the healthy culture of *M. amblycepha*, as well as provide some theoretical basis for nutrient metabolism through the intestine–liver axis. However, further validation of the target metabolite mechanism of *Lactobacillus plantarum* in regulating starch metabolism is still needed.

## Figures and Tables

**Figure 1 animals-15-00583-f001:**
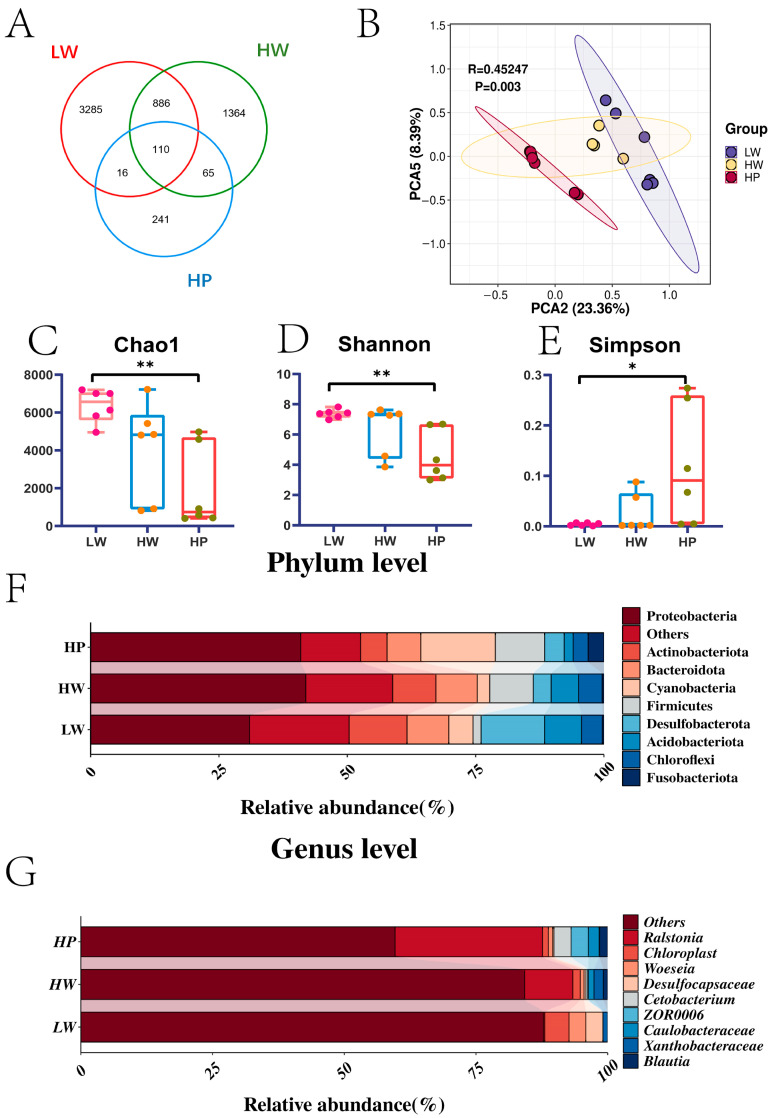
*Lactobacillus plantarum* modulated gut microbiota composition. Note: (**A**) OTU Venn diagram. Red represented the LW group, green represented the HW group, and blue represented the HP group. (**B**) PCA analysis. Purple represented the LW group, yellow represented the HW group, and red represented the HP group. (**C**–**E**) Alpha diversity. The top, middle and bottom horizontal lines indicated the maximum, average, and minimum points. The pink box represented the LW group, the blue box represented the HW group, and the red box represented the HP group. (**F**) The heatmap of gut microbiota abundance at the phylum level. (**G**) The heatmap of gut microbiota abundance at the genus level. * represented *p* < 0.05, ** represented *p* < 0.01 (independent *t* test).

**Figure 2 animals-15-00583-f002:**
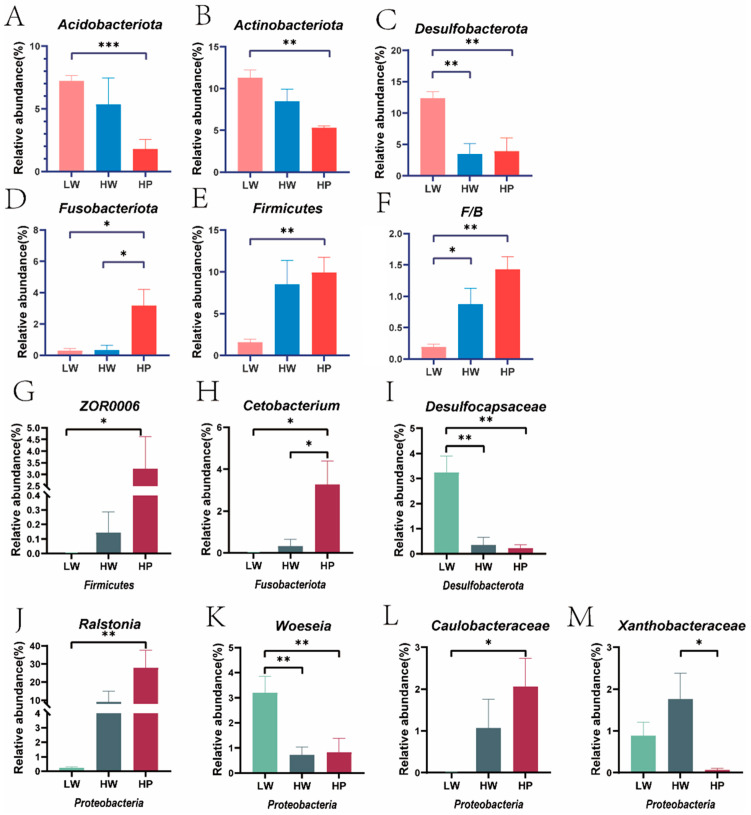
Microbes that differed at the phylum and genus levels. Note: (**A**–**E**) the differences in gut microbiota at the phylum level. Pink represented the LW group, blue represented the HW group, and red represented the HP group. (**F**) The ratio of Firmicutes to Bacteroidota. (**G**–**M**) The differences in gut microbiota at the genus level. Green represented the LW group, gray represented the HW group, and red represented the HP group. * represented *p* < 0.05, ** represented *p* < 0.01, *** represented *p* < 0.001 (Independent *t* test).

**Figure 3 animals-15-00583-f003:**
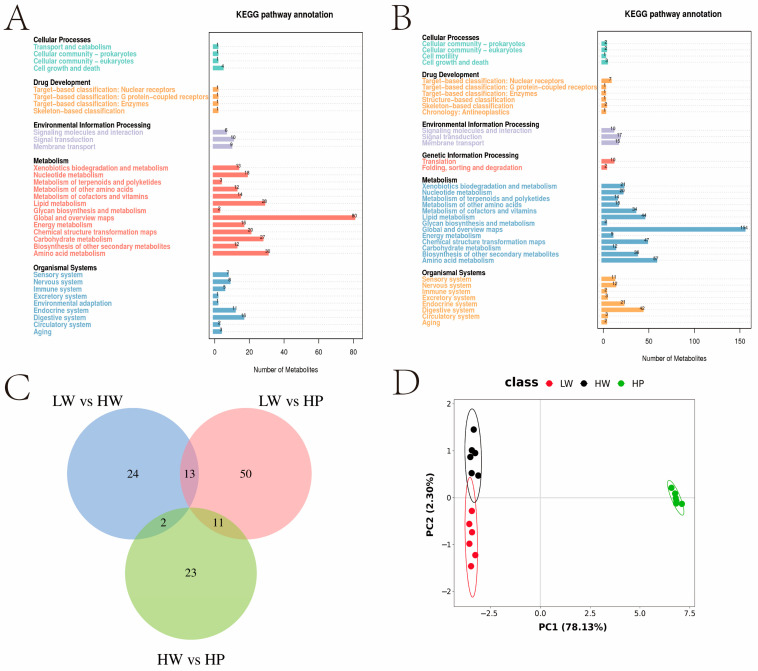
*Lactobacillus plantarum* modulated metabolite composition. Note: (**A**) metabolites annotated to the KEGG database in positive ion mode. (**B**) Metabolites annotated to the KEGG database in negative ion mode. (**C**) Venn diagrams of metabolite differences between the two groups. Blue represented differential metabolites in the LW and HW groups, pink represented differential metabolites in the LW and HP groups, and green represented differential metabolites in the HW and HP groups. (**D**) Principal component analysis of three groups. Red represented data in the LW group, black represented data in the HW group, and green represented data in the HP group.

**Figure 4 animals-15-00583-f004:**
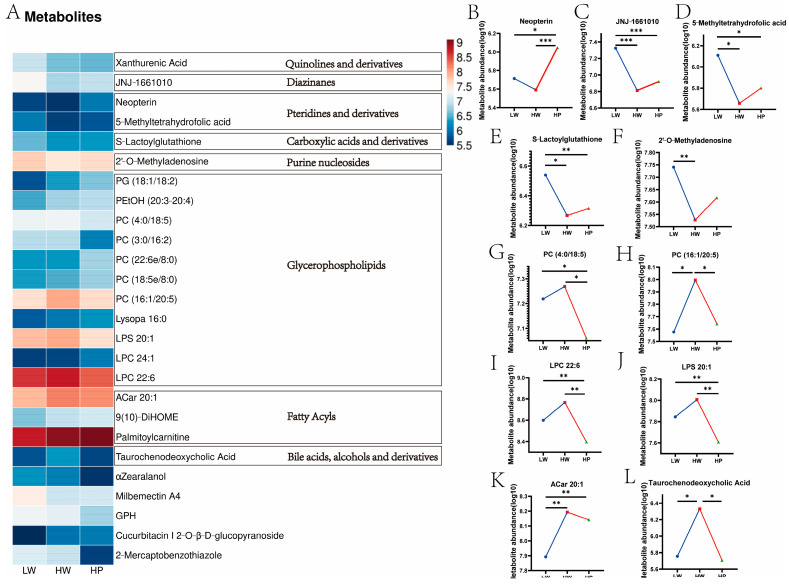
Metabolites analyses with significant differences. Note: (**A**) heatmap of shared differential metabolites. Data were processed for log10. (**B**–**L**) Differential metabolites. Blue dots indicated the LW group data, red boxes indicated the HW group data, green triangles indicated the HP group data. * represented *p* < 0.05, ** represented *p* < 0.01, *** represented *p* < 0.001 (independent *t* test).

**Figure 5 animals-15-00583-f005:**
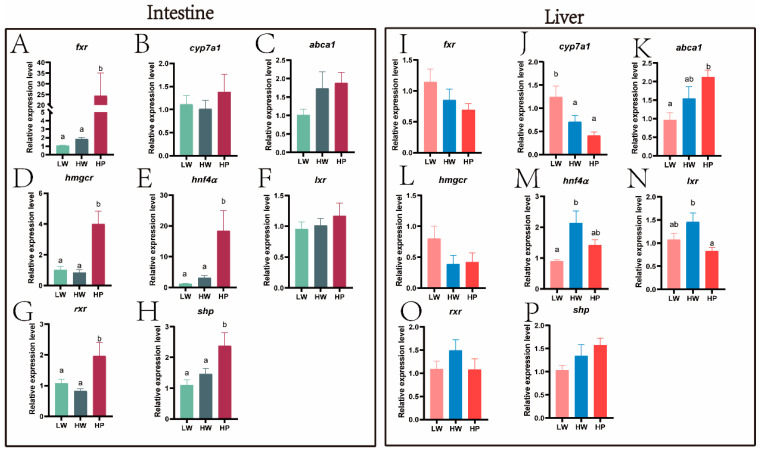
Relative expression of bile acid metabolism genes. Note: Gene relative expressions related to bile acids metabolism in the intestine (**A**–**H**) and in the liver (**I**–**P**). Green represented the LW group, gray represented the HW group, and red represented the HP group (**A**–**H**). Pink represented the LW group, blue represented the HW group, and red represented the HP group (**I**–**P**). Different letters indicated significant differences between groups (*p* < 0.05, Duncan’s test).

**Figure 6 animals-15-00583-f006:**
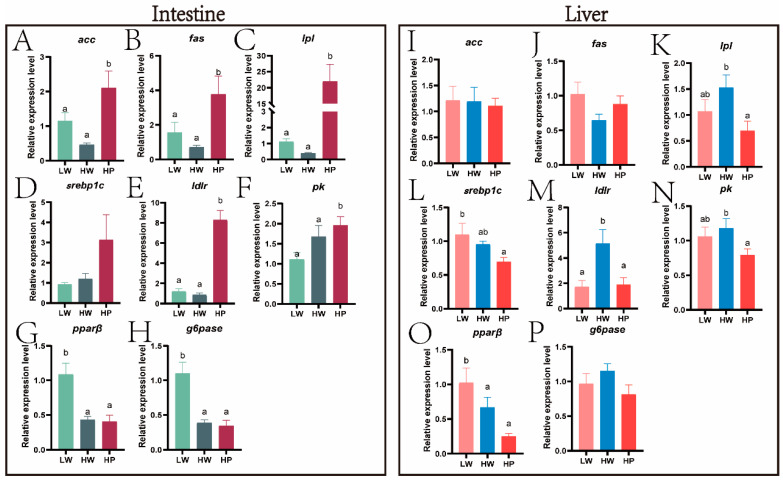
Relative expression of glucolipid metabolism genes. Note: Gene relative expressions related to glycolipid metabolism in the intestine (**A**–**H**) and in the liver (**I**–**P**). Green represented the LW group, gray represented the HW group, and red represented the HP group (**A**–**H**). Pink represented the LW group, blue represented the HW group, and red represented the HP group (**I**–**P**). Different letters indicated significant differences between groups (*p* < 0.05, Duncan’s test).

**Figure 7 animals-15-00583-f007:**
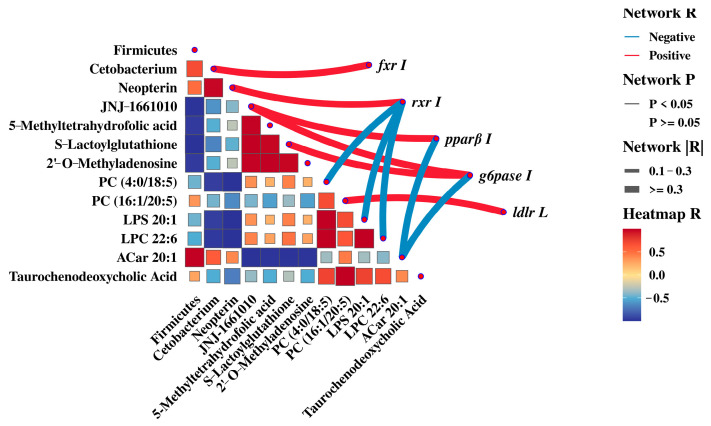
Gut microbiota, metabolites, and genes correlation analyses. Note: the heatmap used Pearson for correlation analysis, with red indicating a positive correlation and blue indicating a negative correlation. Darker colors and larger squares indicated higher correlation. Network diagrams used correlation analysis by correlation test. The solid line indicated *p* < 0.05. The thickness of the line represented the size of the absolute value of the correlation.

**Figure 8 animals-15-00583-f008:**
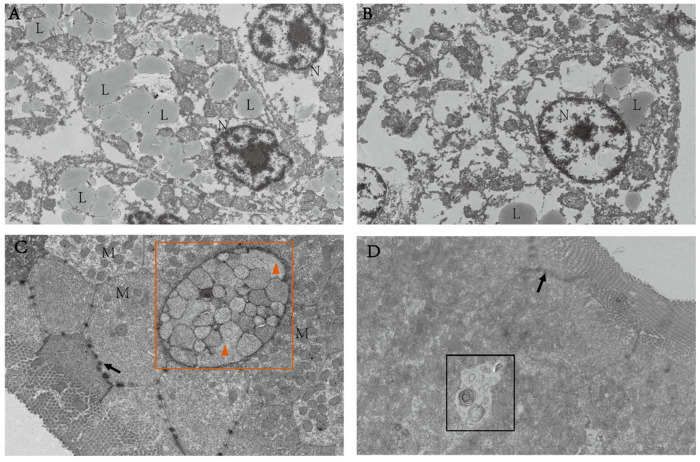
The histopathological responses of the liver and intestine in the HW and HP groups were evaluated using electron microscope section analysis. Note: (**A**) electron microscope section of liver in the HW group (1 µm, the same as blow). (**B**) Electron microscope section of liver in the HP group. (**C**) Electron microscopic section of intestine in the HW group. (**D**) Electron microscope section of intestine in the HP group. L: lipid droplets; N: nucleus; M: mitochondrion. The black arrow represented the bridges. Orange box indicated secretory vesicle, and triangles indicated the appearance of fusion of mucus. Black box indicated the appearance of autophagy.

**Table 1 animals-15-00583-t001:** Ingredient and nutrient composition of the experimental diets.

Ingredient (%)	LW	HW
Fishmeal ^1^	6.4	6.4
Soybean meal (46%) ^1^	25.4	21.6
Rapeseed meal ^1^	16.2	16.2
Cottonseed meal (50%) ^1^	15.3	15.3
Wheat meal ^1^	16.6	33.2
Rice bran	5.7	0.0
Bran	5.8	0.0
Soybean oil ^2^	3.1	3.9
Monocalcium phosphate	1.0	1.0
Premix ^1^	1.0	1.0
Vitamin C ^1^	0.5	0.5
Choline chloride ^1^	0.4	0.4
Microcrystalline cellulose ^1^	2.1	0.0
Bentonite ^1^	0.5	0.5
Total	100	100
Nutrient composition, % air-dried base		
Crude protein	33.49	32.66
Crude fat	6.23	6.57
Ash	7.9	7.3
Gross energy (KJ/g) ^3^	17.48	17.49

Note: ^1^ Provided by Dabeinong Group Huaian Branch (Huaian, China). Fish meal, crude protein 613 g/kg, crude lipid 102 g/kg, selenite 4.70 mg/kg; soybean meal, crude protein 441.7 g/kg, crude lipid 11 g/kg, selenite 0.13 mg/kg; rapeseed meal, crude protein 375.4 g/kg, crude lipid 14 g/kg, selenite 1.10 mg/kg; cottonseed meal, crude protein 493.2 g/kg, crude lipid 14 g/kg, selenite 0.27 mg/kg; wheat meal, crude protein 110 g/kg, crude lipid 12 g/kg. Mineral premix composition (g/kg of premix): calcium diphosphate, 20 g; sodium chloride, 2.6 g; potassium chloride, 5 g; magnesium sulfate, 2 g; ferrous sulfate, 0.9 g; zinc sulfate, 0.06 g; cupric sulfate, 0.02 g; manganese sulfate, 0.03 g; cobalt chloride, 0.05 g; potassium iodide, 0.004 g. ^2^ Provided by Fulinmen Commodity Soybean Oil, purchased from the local RT-mart supermarket (Wuxi, China). ^3^ Total energy was calculated from the energy coefficients of protein, fat and carbohydrate (23.6, 39.5 and 17.2 kJ/g, respectively).

**Table 2 animals-15-00583-t002:** Primers sequences for qPCR.

Gene	Forward Primer (5′-3′)	Reverse Primer (5′-3′)	Genbank No.	Product Length (bps)
*fxr*	TGGTGGGGAGGATTGGTACT	CTACAGGGCAAGACTCGTCG	XM_048169656.1	101
*ldlr*	ACTGATGACTGTGGAGACGG	AGGTCTTAGGCACACACTGG	XM_048158987.1	115
*lxr*	GCACGTACCTCTACAGTGGC	GAGCGTTTGTTGCACTGCTT	XM_048184573.1	119
*cyp7a1*	TTTCCGTCAGACGCTTCAGG	CCCTTCTTCAAGCCAGTCGT	XM_048186424.1	118
*rxr*	GCCATATTCGACAGGGTGCT	ACTCCACTTCACTTGGGCTG	XM_048202891.1	142
*hmgcr*	CATGTGCTCTGGCCAAGTTT	AGTTAGCCAGGACAGACAAACC	XM_048189198.1	204
*hnf4α*	GGGGAGCATATCCCAATGCC	ATGTGTGAAGACCTCAGCCC	XM_048186071.1	111
*abca1*	AGGACTACTCGGTGTCCCAG	ACCGCTGTCTCTTTACGACG	XM_048177071.1	109
*shp*	GGGCAGCATCCCAACTGTAA	CGCAGGACTTCGTCACCTTT	XM_048153777.1	149
*srebp1c*	ACAACAGTAGCGACACCCTG	CATCAGTGGAACGGTGGTCA	XM_048187188.1	117
*pk*	GCCGAGAAAGTCTTCATCGCGCAG	CGTCCAGAACCGCATTAGCCAC	XM_048152870.1	157
*g6pase*	TTCAGTGTCACGCTGTTCCT	TCTGGACTGACGCACCATTT	XM_048171060.1	119
*acc*	TAGCAGTGAGCATTGGCACA	CATCGCTGGCGTATGAGGAT	XM_048188636.1	327
*fas*	GTTTGCCAACCGCTTGTCTT	GGCCATGGCGAATAGCATTG	XM_048171583.1	119
*lpl*	TCTGATGGGATCTGGCAC	GTTTCTGGATTTGGGTCG	XM_048164066.1	85
*pparβ*	CATCCTCACGGGCAAGAC	CACTGGCAGCGGTAGAAG	XM_048209548.1	153
*β-actin*	TCGTCCACCGCAAATGCTTCTA	CCGTCACCTTCACCGTTCCAGT	AY170122.2	152

## Data Availability

The raw 16S rRNA sequencing data can be accessed at Sequence Read Archive (https://www.ncbi.nlm.nih.gov/sra) (accessed on 20 July 2024) using the accession number PRJNA1130113.

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
