# Peer review of "Lactobacillus plantarum* Alters Gut Microbiota and Metabolites Composition to Improve High Starch Metabolism in *Megalobrama amblycephala"

_animals, 2025, doi:10.3390/ani15040583_

Round 1
Reviewer 1 Report
Comments and Suggestions for Authors
1. The introduction section comprehensively presents the background by encompassing the key concepts relevant to this study and citing relevant research to substantiate the claims.
2. The experimental design is sound. It incorporates a control group and experimental groups, an appropriate sample size, a sufficient feeding period, and multi-faceted data collection methods, facilitating a comprehensive assessment of the impact of Lactobacillus plantarum on bighead carp fed a high-starch diet.
3. The research results are clearly and precisely presented through the utilization of appropriate tables, figures, and detailed textual descriptions, effectively communicating the findings.
4. The conclusions are fundamentally validated by the obtained results.
a. The gut microbiota analysis results indicate that, in contrast to the LW group, the ratio of Firmicutes/Bacteroidota and the abundance of Cetobacterium in the HP group exhibited a significant increase. This finding lends support to the conclusion that Lactobacillus plantarum can regulate the gut microbiota disorders induced by a high-starch diet.
b. The metabolite analysis outcomes reveal that the levels of PC(16:1/20:5) and taurochenodeoxycholic acid in the HP group were notably lower than those in the HW group. This corroborates the conclusion that Lactobacillus plantarum can modify the metabolism of carboxylic acids and their derivatives as well as glycerophospholipid-related metabolites.
c. The gene expression analysis demonstrates that the expression of genes related to the bile acid cycle (fxr, hmgcr, rxr, shp, and hnf4α) in the intestine of the HP group was significantly higher than that in the LW and HW groups. Moreover, the expression of genes associated with glycolysis (acc, fas, lpl, ldlr, and pk) in the intestine of the HP group was also significantly higher than that in the other two groups. These results provide strong evidence in support of the conclusion that Lactobacillus plantarum can activate intestinal glycolysis and stimulate the bile acid cycle.
d. The transmission electron microscopy analysis discloses that, compared with the HW group, the area of lipid droplets in the liver of the HP group was significantly reduced, and autophagy was observed in the intestine. These observations firmly support the conclusion that Lactobacillus plantarum can mitigate hepatic lipid deposition and induce mitochondrial autophagy.
Suggestions for Revision:
1. The conclusion regarding the reduction of gut microbial diversity upon adding LAB to a high-starch diet, potentially due to the competitive elimination of pathogenic bacteria, requires further evidence. A more in-depth discussion or a refinement of the statement is necessary to enhance its credibility.
2. In numerous similar studies on LAB, the influence of the addition amount has been considered. It is essential to clarify the rationale behind the selected addition amount in this study and explain the absence of an addition amount gradient in the experimental design.
3. In the practical application of LAB in aquaculture, there are distinctions between dead and live bacteria. The author is expected to discuss the justification for using live bacteria in this study and comprehensively analyze the advantages and disadvantages of live bacteria.
4. The language quality is generally acceptable; however, there are some minor grammatical errors and less-than-optimal expressions that could slightly impede the understanding of the study. It is recommended to seek the assistance of a native English speaker or an expert in academic writing to meticulously revise the text.
Concerning the interpretation of the results, despite their clear presentation, a more profound and comprehensive analysis is warranted. The authors ought to engage in a more detailed discussion regarding the implications of the results and their correlations with the extant literature.
With respect to the scope of the conclusions, while they are in harmony with the obtained results, they could be more far-reaching. The authors are encouraged to explore the broader ramifications of their research findings and to propose potential avenues for future research endeavors.
Author Response
Dear Editor,
Thank you for allowing us to revise our manuscript. We also appreciate the constructive and professional comments from the reviewers (animals-3422958). Please check our point-to-point responses to the comments and the corresponding revisions in the revised manuscript (highlighted in red).
I hope we have adequately addressed the comments and suggestions from reviewers. Please feel free to contact me if any further improvements to the manuscript are deemed necessary.
Best wishes
Linghong Miao, miaolh@ffrc.cn
Wuxi Fisheries College, Nanjing Agricultural University
Wuxi Shanshui East Road No.9, Jiangsu, China
The first author: Linjie Qian, qianlinjiejie@gmail.com
Wuxi Fisheries College, Nanjing Agricultural University
Wuxi Shanshui East Road No.9, Jiangsu, China
Comments 1:
Suggestions for Revision:
- The conclusion regarding the reduction of gut microbial diversity upon adding LAB to a high-starch diet, potentially due to the competitive elimination of pathogenic bacteria, requires further evidence. A more in-depth discussion or a refinement of the statement is necessary to enhance its credibility.
Response: Thanks for your advice. The decreasing trend of gut microbial diversity was found in Oncorhynchus mykiss on the 29th day after feeding commercial diets with Bacillus subtilis. (Hines et al., 2022). In dairy cows, the Lactobacillus casei Zhang and Lactobacillus plantarum P-8 treatment group was found to significantly increase the abundance of beneficial bacteria but suppress some conditionally pathogenic bacteria (Xu et al., 2017). The treatment of patients with non-alcoholic mild hepatic encephalopathy by Bacillus subtilis and Enterococcus faecium reduced the diversity of gut microbiota and decreased the abundance of ammonia-producing microbiota belonging to Firmicutes (Zuo et al., 2017). The use of Bifidobacterium bifidum to treat H. pylori-positive patients showed a dramatic decrease in alpha diversity and a reduction in H. pylori (Peng et al., 2023). Therefore, we hypothesized that the decrease in gut microbial diversity in the LAB to high-starch diet could be due to competitive elimination of pathogenic bacteria (highlighted in RED, Line383-388).
Reference:
Hines I S, Santiago-Morales K D, Ferguson C S, et al. Steelhead trout (Oncorhynchus mykiss) fed probiotic during the earliest developmental stages have enhanced growth rates and intestinal microbiome bacterial diversity [J]. Frontiers in Marine Science, 2022, 9: 1021647. https://doi.org/10.3389/fmars.2022.1021647
Xu H, Huang W, Hou Q, et al. The effects of probiotics administration on the milk production, milk components and fecal bacteria microbiota of dairy cows[J]. Science Bulletin, 2017, 62(11): 767-774. https://doi.org/10.1016/j.scib.2017.04.019
Zuo Z, Fan H, Tang X D, et al. Effect of different treatments and alcohol addiction on gut microbiota in minimal hepatic encephalopathy patients[J]. Experimental and Therapeutic Medicine, 2017, 14(5): 4887-4895. https://doi.org/10.3892/etm.2017.5141
Peng R, Zhang Z, Qu Y, et al. The impact of Helicobacter pylori eradication with vonoprazan-amoxicillin dual therapy combined with probiotics on oral microbiota: a randomized double-blind placebo-controlled trial[J]. Frontiers in Microbiology, 2023, 14: 1273709. https://doi.org/10.3389/fmicb.2023.1273709
- In numerous similar studies on LAB, the influence of the addition amount has been considered. It is essential to clarify the rationale behind the selected addition amount in this study and explain the absence of an addition amount gradient in the experimental design.
Response:Thanks for your kind reminder. According to the literature we selected to add LAB with theoretical 108 CFU/g (Van et al., 2017; Li et al., 2022) and 109 CFU/g (Sliva et al., 2021). However, after plate scribing of the feed it was determined that the actual probiotic concentration in the feed was 106 and 107 CFU/g. The addition of 107 CFU/g of feed did not significantly improve growth performance. Therefore, we chose 106 CFU/g for this study.
Reference:
Van Doan H, Hoseinifar S H, Dawood M A O, et al. Effects of Cordyceps militaris spent mushroom substrate and Lactobacillus plantarum on mucosal, serum immunology and growth performance of Nile tilapia (Oreochromis niloticus) [J]. Fish & shellfish immunology, 2017, 70: 87-94. https://doi.org/10.1016/j.fsi.2017.09.002
Li S, Guo L, Si X, et al. Lactobacillus plantarum WCFS1 alleviates Aeromonas hydrophila NJ-1-induced inflammation and muscle loss in zebrafish (Danio rerio) [J]. Aquaculture, 2022, 548: 737603. https://doi.org/10.1016/j.aquaculture.2021.737603
Silva V V, Salomão R A S, Mareco E A, et al. Probiotic additive affects muscle growth of Nile tilapia (Oreochromis niloticus) [J]. Aquaculture Research, 2021, 52(5): 2061-2069. https://doi.org/10.1111/are.15057
- In the practical application of LAB in aquaculture, there are distinctions between dead and live bacteria. The author is expected to discuss the justification for using live bacteria in this study and comprehensively analyze the advantages and disadvantages of live bacteria.
Response:Thanks for your suggestion. Most probiotics required a sufficient number of live bacteria to be effective. Probiotics may be killed by stomach acid or high concentration of bile salts as they pass through the digestive tract (Zhang X, 2015). LAB was a normal flora in the fish intestine, which could inhibit the growth of pathogenic bacteria by metabolizing organic acids, bacteriocins, hydrogen peroxide and other substances (Wang et al., 2023). Both live and heat-inactivated bacteria of LAB were able to adhere to IPEC-J2 cells, but the competition and displacement inhibition of Escherichia coli adherence to IPEC-J2 cells by the live bacteria was superior to that of the heat-sterilized bacteria (Zeng et al., 2017). After entering the digestive tract of fish, the live bacteria can inhibit the growth of harmful bacteria in the intestinal tract through growth and reproduction, the number of which increases sharply, thus regulating the micro-ecological balance of the fish body (Wang, 2020). (highlighted in RED, Line70-73).
Reference:
Zhang X. Optimization of Culture Conditions and Efficacy Evaluation on Composite Probiotics Microecological Additive [D]. Jilin University,2015.
Wang M, Lv C, Yang X et al., Effects of Lactobacillus plantarum (LP HMX-3) on growth, digestion, immunity and intestinal flora of Apostichopus japonicus [J]. Journal of Fisheries of China, 2023,47(12):137-148. https://doi.org/10.11964/jfc.20210913081
Zeng Y, Ji H, Wang S et al., Effect of live and heat-inactivated Lactobaillus plantarum to Escherichia coli adhesion on IPEC-J2 cell [J]. Journal of Gansu Agricultural University, 2017,52(06):11-17. https://doi.org/10.13432/j.cnki.jgsau.2017.06.003.
Wang Z. Application of probiotics in aquatic animal cultivation [J]. Henan Shuichan, 2020,(05):10-12.
- The language quality is generally acceptable; however, there are some minor grammatical errors and less-than-optimal expressions that could slightly impede the understanding of the study. It is recommended to seek the assistance of a native English speaker or an expert in academic writing to meticulously revise the text.
Response: Thanks for your kind reminder. We have corrected the grammatical errors and expressions in the manuscript.
Concerning the interpretation of the results, despite their clear presentation, a more profound and comprehensive analysis is warranted. The authors ought to engage in a more detailed discussion regarding the implications of the results and their correlations with the extant literature.
Response: Thanks for the advice. We discussed them in more detail in the context of the existing literature, and they were highlighted in red in the manuscript.
With respect to the scope of the conclusions, while they are in harmony with the obtained results, they could be more far-reaching. The authors are encouraged to explore the broader ramifications of their research findings and to propose potential avenues for future research endeavors.
Response: Thanks for your suggestion. The conclusions have been expanded (highlighted in RED, Line 491-499).
Reviewer 2 Report
Comments and Suggestions for Authors
My comment to the paper as below:
1. Title: accepted
but may consider title as below:
Lactobacillus plantarum Alters Gut Microbiota and 2 Metabolites Composition in Megalobrama amblycephala fed high starch diet
2. Abstract:
How many replicate per treatment?
the net is only 1m X 1m X 1m? look like too small
Line 27: why need double bracket?
3. introduction
too many old citations is not good. Try to replace with the latest references
provide a paragraph the need of this study. any research gap? problem statement?
4. Materials and methods
Provide information on how to measure water quality. How the determine photoperiod is 12 d 12 n?
Any references to support your experiment in 2.5?
Authors need to clarify HOV and data normality test before running anova test
5. conclusion
need to put more efforts in this section by providing suggestion in which dose of the LAb can be applied in the fish farming.
what the research gap and future work?
please check scientific name format - check for whole manuscript
Looking forward your revision
Author Response
Dear Editor,
Thank you for allowing us to revise our manuscript. We also appreciate the constructive and professional comments from the reviewers (animals-3422958). Please check our point-to-point responses to the comments and the corresponding revisions in the revised manuscript (highlighted in red).
I hope we have adequately addressed the comments and suggestions from reviewers. Please feel free to contact me if any further improvements to the manuscript are deemed necessary.
Best wishes
Linghong Miao, miaolh@ffrc.cn
Wuxi Fisheries College, Nanjing Agricultural University
Wuxi Shanshui East Road No.9, Jiangsu, China
The first author: Linjie Qian, qianlinjiejie@gmail.com
Wuxi Fisheries College, Nanjing Agricultural University
Wuxi Shanshui East Road No.9, Jiangsu, China
Comments 2:
My comment to the paper as below:
- Title: accepted
but may consider title as below:
Lactobacillus plantarum Alters Gut Microbiota and 2 Metabolites Composition in Megalobrama amblycephala fed high starch diet
Response: Thanks for your advice. According to your comment, we have revised the title. Lactobacillus plantarum Alters Gut Microbiota and Metabolites Composition to improve High Starch Metabolism in Megalobrama amblycephala (highlighted in RED, Line 2-4).
- Abstract:
How many replicate per treatment?
Response: Thanks for your advice. We set up three replicates for each treatment.
the net is only 1m X 1m X 1m? look like too small
Response: Thanks for your comments. Based on previous research (Wang Y, 2019), it was found that the growth performance of M.amblycephala increased significantly when stocking density was less than 60 fish/m3 (initial mean weight of 8.20 ± 0.01g), while it decreased when stocking density was greater than 60 fish/m3. In our study, 20 juvenile M.amblycephala (initial mean weight of 13.5 ± 0.5g) were stocked in each net, which was suitable for the growth of M.amblycephala.
Reference:
Wang Y. Effects of hypoxia and culture density on physiological and biochemical indexes and tissue structure of Megalobrama amblycephala [D]. Shanghai Ocean University, 2019. https://doi.org/10.27314/d.cnki.gsscu.2019.000141
Line 27: why need double bracket?
Response: Thanks for your kind reminder. We have changed this in the manuscript to (13.5±0.5g) (highlighted in RED, Line 24).
- introduction
too many old citations is not good. Try to replace with the latest references
Response: Thank you for your advice. We searched for relevant literature in recent years and replaced them in the manuscript (highlighted in RED, Line 44-47, 49-55, 75-78).
provide a paragraph the need of this study. any research gap? problem statement?
Response: Thank you for your suggestion. As we mentioned in Line 89-92, there was currently limited information on how LAB regulates the interaction between glucose metabolism and bile acid metabolism through the modulation of gut microbiota and metabolite composition in M.amblycephala. We hope to find the key gut microbiota and metabolites of LAB that regulate glucose-lipid metabolism in this study, thereby improving the utilization of high starch. We further investigated the key gut microbiota and metabolites that regulate high starch metabolism, and explored how LAB impacts the regulatory targets of high starch metabolism through the intestine, which will provide an important basis for the healthy breeding of M.amblycephala (highlighted in RED, Line 101-104).
- Materials and methods
Provide information on how to measure water quality. How the determine photoperiod is 12 d 12 n?
Response: Thanks for the suggestion. We used the national standard method (HJ/T 195-2023) to measure ammonia nitrogen concentration in the water environment. Dissolved oxygen and temperature in the waters were monitored three times a day using a dissolved oxygen meter and thermometer. Our breeding site was located at 120.29E, 31.43N. During the breeding period, the sun rose at about 5:30 and set at 18:30. We counted the natural light as the sun appearing completely on the horizon. Therefore, the light time is from 6:00 to 18:00. Therefore, we considered the natural photoperiod to be 12d:12n.
Any references to support your experiment in 2.5?
Response: Thank you for your advice. We recommend to refer to previous literature in our lab for sample collection (Jing et al.,2020; Liu et al., 2024; Jiang et al., 2024).
Reference:
Jing H, Yan L, Wen P, et al. Dietary selenium enhances the growth and anti-oxidant capacity of juvenile blunt snout bream (Megalobrama amblycephala) [J]. Fish & Shellfish Immunology, 2020, 101. https:// doi.org/10.1016/j.fsi.2020.03.041
Liu H, Gu Z, Lin Y, et al. Activated charcoal supplementation in cottonseed meal-based feed improved growth performance and antioxidant capacity through enhancing intestinal barrier function in grass carp juveniles (Ctenopharyngodon idellus) [J]. Aquaculture Reports, 2024, 39: 102442. https://doi.org/10.1016/j.aqrep.2024.102442
Jiang W, Qian L, Mu Q, et al. Endoplasmic reticulum stress and Ca2+ dysregulation in response to ammonia nitrogen exposure could be alleviated by dietary fermented mulberry leaf meal in Megalobrama amblycephala [J]. Aquaculture, 2024: 741256. https://doi.org/10.1016/j.aquaculture.2024.741256
Authors need to clarify HOV and data normality test before running anova test
Response: Thanks for your advice. We checked the normal distribution of the data before performing a one-way ANOVA or independent T-test (highlighted in RED, Line148-149).
- conclusion
need to put more efforts in this section by providing suggestion in which dose of the LAb can be applied in the fish farming.
Response: Thank you for your suggestion. Before this experiment we used two concentrations (108 CFU/g and109 CFU/g) of LAB based on literature (Van et al., 2017; Li et al., 2022; Sliva et al., 2021). However, after plate scribing of the feed it was determined that the actual probiotic concentration in the feed was 106 and 107 CFU/g. The previous growth results showed that 106CFU/g was able to significantly improve the growth performance while 107CFU/g showed no significant difference. Therefore, we conclude that 106CFU/g is more favorable for high glucose metabolism for growth in Megalobrama amblycephala based on the growth performance and the results of this experiment.
Reference:
Van Doan H, Hoseinifar S H, Dawood M A O, et al. Effects of Cordyceps militaris spent mushroom substrate and Lactobacillus plantarum on mucosal, serum immunology and growth performance of Nile tilapia (Oreochromis niloticus) [J]. Fish & shellfish immunology, 2017, 70: 87-94. https://doi.org/10.1016/j.fsi.2017.09.002
Li S, Guo L, Si X, et al. Lactobacillus plantarum WCFS1 alleviates Aeromonas hydrophila NJ-1-induced inflammation and muscle loss in zebrafish (Danio rerio) [J]. Aquaculture, 2022, 548: 737603. https://doi.org/10.1016/j.aquaculture.2021.737603
Silva V V, Salomão R A S, Mareco E A, et al. Probiotic additive affects muscle growth of Nile tilapia (Oreochromis niloticus) [J]. Aquaculture Research, 2021, 52(5): 2061-2069. https://doi.org/10.1111/are.15057
what the research gap and future work?
Response: Thanks for the suggestion. The conclusions have been expanded (highlighted in RED, Line 491-499).
please check scientific name format - check for whole manuscript
Response: Thanks for your comments. We have carefully checked all scientific name in the text. If there were errors, the corrected scientific names were highlighted in red in the manuscript.